# New Species of *Paussus*, Subgenus *Scaphipaussus* (Coleoptera: Carabidae: Paussinae), from Southeast Asia Reveal Ambiguities in Species Group Limits and High Species Diversity in the Oriental Region [note 1]

**DOI:** 10.3390/insects14120947

**Published:** 2023-12-14

**Authors:** Michal Bednařík, Ladislav Bocak

**Affiliations:** 1Independent Researcher, Tř. Spojenců 10, 779 00 Olomouc, Czech Republic; 2Czech Advanced Science and Technology Institute, Palacký University, Slechtitelu 27, 779 00 Olomouc, Czech Republic; ladislav.bocak@upol.cz

**Keywords:** ground beetles, ant nest beetles, new species, taxonomy, zoogeography

## Abstract

**Simple Summary:**

We studied the species diversity of Asian *Paussus*, specifically focusing on the subgenus *Scaphipaussus*. These extraordinary and highly modified ground beetles (Coleoptera: Carabidae) live within ant nests and are seldom collected. Our investigation led to the discovery of nine new species within this group, and we document their high phenotypic divergence. The latest classification of *Paussus* was based on molecular phylogeny. Still, due to their scarcity, obtaining DNA samples is a challenge. Therefore, at least provisionally, further morphological data help with the classification of species groups and identification of the centers of diversity in Southeast Asia. Over the past decade, ~30 *Scaphipaussus* species have been described by different researchers, hinting that there might be many more waiting to be discovered. Further research should use preserved specimens for genetic studies. This will not only help us understand their evolutionary origins but also shed light on the intricate relationships between these beetles and their ant hosts.

**Abstract:**

*Paussus*, commonly known as ant nest beetles, is the most diverse genus of Paussinae (Coleoptera: Carabidae) with a very complex taxonomic history. Biodiversity research in Southeast and South Asia yields new species that can contribute to a better understanding of the morphological disparity and species-group or subgenus delimitation. Here, we describe nine new species from Southeast Asia and China: *Paussus* (*Scaphipaussus*) *fencli* sp. nov. (China), *P*. (*S*.) *mawdsleyi* sp. nov. (Borneo), *P*. (*S*.) *bakeri* sp. nov. (Philippines), *P*. (*S*.) *jendeki* sp. nov. (Laos), *P*. (*S*.) *saueri* sp. nov. (India), *P*. (*S*.) *annamensis* sp. nov. (Vietnam), *P*. (*S*.) *phoupanensis* sp. nov. (Laos, Vietnam), *P*. (*S*.) *bilyi* sp. nov. (Thailand), and *P*. (*S*.) *haucki* sp. nov. (Thailand). We also bring new data on *P*. (*S*.) *corporaali* Reichensperger, 1927 (Java) and *P*. (*S*.) *madurensis* Wasmann, 1913 (India). Besides formal descriptions, we provide photographs of the habitus in the dorsal and dorsolateral view, antennal club, head crest, and male genitalia if the male is available. Based on the comparison of new and earlier described species, we show that the antennae are highly diverse within the *Scaphipaussus*. Considering other characters, some species are placed in *Scaphipaussus*, but they differ from putative relatives in the antennal morphology. The presence of the frontal protuberances and crests is a more reliable character. Additional species show that *Scaphipaussus* is most diverse in southeastern Asia, especially in Indo-Burma. Concerning its supposed late Miocene origin, the group underwent rapid radiation. The species diversity of *Scaphipaussus* almost doubled in the last decade, and it is highly probable that further species will be described in the future.

## 1. Introduction

The charismatic and ecologically interesting ant nest beetle genus *Paussus* Linnaeus, 1775 represents a highly diverse clade of the tribe Paussini (Carabidae: Paussinae), at present, with over 380 spp. [1,2,3,4,5,6,7,8]. The phenotypic distinctiveness prompted the earlier taxonomists to give these modified ground beetles a high rank [9,10]. Already morphology-based studies included the paussines in ground beetles [11,12]. Analogically, the related rhysodines (Carabidae: Rhyssodinae) that have different biology and disparate morphology akin to Paussinae are now included in the ground beetles [9,13,14]. Molecular research, ranging from short fragment analyses to comprehensive phylogenomic studies, has consistently supported these relationships [13,15,16,17].

*Paussus* is not only the type-genus, but it also represents the largest radiation in the subfamily. It belongs to the most modified paussine beetles, and its taxonomy has a very convoluted history. The genus in the current sense [3,5,18,19,20] had been earlier split into many subgenera, some being occasionally given a full genus rank [1,21,22]. The backbone of the *Paussus* classification was solved by the latest morphology- and DNA-based classification [3] (Figure 1A). The authors analyzed DNA data of 57 spp. (~15% of formally described species diversity) and defined three subclades in *Paussus*, designated as series I–III. They also lowered the number of accepted subgenera based on widely encountered polyphyly of several morphology-based taxa. Simultaneously, their detailed analysis showed that some groups could not be clearly defined, and some species included in the analyses had to remain in a *Paussus incertae sedis* position (Figure 1A). The placement of species in subgenera was mainly based on molecular evidence. The detailed comparison of morphology- and DNA-based groups was beyond the project’s scope, as properly preserved material for DNA analyses is scarce. It is worth noting that only two species of *Cochliopaussus* and one *Semipaussus* were recovered in the *Scaphipausssus* clade by Robertson & Moore [3], and the respective subgenera were synonymized [3]. Their results showed that morphology alone can sometimes be misleading, and the classification needs further attention. Therefore, the authors presented in the Supplements the actual overview of valid subgenera and species. Most species were assigned to the series I–III and subgenera, but some *Paussus* species earlier belonging to the polyphyletic and synonymized subgenera became unassigned [3]. Therefore, the data on further species are essential for gradually building the natural classification and a better understanding of morphological diversity and distribution of subgenera.

Here, we focus on Oriental *Paussus*, presented as a subgenus *Scaphipaussus* Fowler, 1912, or the *P. hystrix* group [3,23]. This subgenus and an informal group belong to the series II defined by Robertson & Moore [3] and approximately corresponds to Darlington’s Paussina series II [22]. The series II contains, besides *Scaphipaussus*, the nominotypical subgenus, the Afrotropical subgenera *Hylotorus* Dalman, 1823, and *Anapaussus* Wasmann, 1929, and additional species left without subgeneric assignment due to their aberrant morphology. The *Paussus* series II is characterized by well-defined temples, a narrow gula, the presence of a frontal crests and/or protuberances [23], an emarginate anterior margin of the labrum, distinctly produced anterolateral angles of ligula, short and robust apical setae of the ligula, wide maxillary palpomere 2, the antennal club with undulate outer (=posterior) margin, flat eyes, the pronotum with an apparent lateral transverse cleft, absent tibial spurs, and a flat or shallowly emarginate apical part of the phallus [3]. *Scaphipaussus* is mainly defined by the molecular phylogeny as the morphological analysis provided lower support (BS 77%) and also includes parts of subgenera *Cochliopaussus* Kolbe, 1927 and *Semipaussus* Wasmann, 1919 [3]. It means that some species have not been studied after these subgenera were synonymized and they might belong to *Scaphipaussus*. The distribution of the earlier described species of *Scaphipaussus* is shown in Figure 1B.

If informal groups are employed, some *Scaphipaussus* species are designated as members of the *P. hystrix* group, including *P*. (*S*.) *waterhousei* Westwood, 1874 [the sometimes still used original spelling “*waterhousii*” is incorrect according to the Code [24], Art. 31.1.2] that is sometimes considered the type species of *Scaphipaussus* [23]. *Scaphipaussus sensu* Robertson & Moore [3] and the *P. hystrix* group *sensu* Maruyama [23] overlap to a great extent but are not identical. The *P. hystrix* group is defined by the clearly defined crest on the vertex, the presence of the short median frontal suture, frontal ridge well demarked but not explanate, rounded margin of the maxillary palpomere II, and the ligula without distinct median keel. Altogether, 27 species have been listed in the *hystrix* species group [23,25,26] out of 76 species in *Scaphipaussus.* Both the group and the subgenus predominantly contain species occurring in the eastern part of the Oriental region (Indo-Burma, Malaya, the Greater Sundas, and the Philippines, Figure 1).

Most *Paussus* occurs in the Old-World tropics, and only a few species reach north of the Palearctic region (North Africa, the Balkan, Levant, Iran, Arabia, Iran, and China, >20 spp.) [3,7]. The group is common in the African savannah ecosystems, and many species also occur in Sub-Saharan tropical forests [1,22]. Although *Paussus* is scarcely collected in the eastern part of the Oriental region, ninety-nine *Paussus*, and among them seventy-six *Scaphipaussus*, have been described from Southern and Eastern Asia, including India, Indo-Burma, Southern China, the Greater Sundas (including the islands on the continental shelf), and the Philippines [4,19,23,25,26,27,28,29,30,31]. The western Oriental fauna shows high subgeneric diversity and apparent relationships to the Afrotropical groups. India is dominated by the majority of species assigned to the Afrotropical groups of *Paussus*, including *Anapaussus* Wasmann, 1929 and *Hylotorus* Dalman, 1823. The Indian species were earlier referenced in *Edaphopaussus* Kolbe, 1920 (8 spp., all from India), and several currently synonymized subgenera, such as *Indupaussus* Luna de Carvalho, 1989 (1 sp. from India and Sri Lanka), *Latipaussus* (1 sp. from Sri Lanka), and *Manicanopaussus* Kolbe, 1929 (2 spp. from India and Sri Lanka). The eastern part of the Oriental region is dominated by *Scaphipaussus* (76 spp., including some species with an uncertain placement). Further, nine east-Oriental species belong to earlier *Curtisipaussus* (at present a subgenus of *Paussus*, *Shuckardipaussus*, and *Paussus incertae sedis*; species from China and the Greater Sundas). One species was placed in earlier *Katapaussus* (now listed in *Paussus incertae sedis*) (a species from Myanmar). A few species have been recorded from the Palearctic region. One *Scaphipaussus* species is known from the Balkan peninsula [4] and seven species from southern China [4,25] (Figure 1). Supposedly, these species colonized the northern areas recently, as shown by the position of *P. turcicus* in the molecular phylogeny [2]. The molecular analyses suggested that the *P. hystrix* group and *Scaphipaussus*, respectively, are relatively young groups that started their radiation about 10 million years ago [2]. It can be therefore expected that the group contains relatively young species that might have evolved during the Pliocene and Pleistocene climatic fluctuations under the conditions of repeated geographic isolation due to dynamic expansion and shrinking of rainforests and isolation by the inundated continental shelf [32].

Although *Paussus* received significant attention from many taxonomists, we can expect that a high proportion of the diversity of this genus remains formally undescribed, as many new species were recently discovered despite the rareness of the group in the collected material [3,23,25,28]. The recent field research of the first author and colleagues yielded further species, and it is worth documenting and naming them and discussing their morphology. The limited knowledge inevitably led to the definition of the complexes consisting of a characteristic species and their closest relatives but with a poorly understood position to other groups. As apparent synapomorphies can be defined for such groups, they were introduced into the classification as genera, recently subgenera, and informal species groups [3,23]. Any expression of relationships in such a large genus is valuable for practical reasons. Still, the constitution of subgenera and species groups, especially their extreme limits, can sometimes be contentious. The concepts can be further defined only through increased taxon sampling and comparing a wide spectrum of related species. Here, we describe new species from a large geographical area from southern India, to Thailand, Laos, Vietnam, southern China, the Philippines, and Borneo (Figure 2). Taxonomic research is crucial for quantifying diversity threatened by anthropogenic pressure. The forest habitats are rendered to urbanized and intensively exploited areas, especially in Southeast Asia. At the same time, taxonomic research is not prioritized in science funding and copes with various obstacles caused by the misunderstanding of science tasks and bureaucracy [33]. We hope that more data can contribute to a better understanding of diversity patterns and phenotypical diversity of paussine beetles, and, ultimately, to the conservation effort.

## 2. Materials and Methods

The study is based on a dry-mounted material collected in the last 30 years by various collectors and some specimens deposited in the major European museums. The museum specimens were collected in historical localities, often in connection with Catholic missions based in China in the early 20th century. The localities can be identified as various plants and animals were described by the earlier authors. The modern names and positions of all sampled localities are shown in Figure 2. The locality data are stated in the labels with only small modifications. The data from different labels are separated by the slash.

The localities with the occurrence of *Paussus* span from highly anthropogenically modified habitats as small fields to less affected forest habitats (Figure 3A,B). Various methods were employed for collecting. The *Paussus* were collected sporadically, mostly in a single or a few specimens, by sweeping the vegetation, usually after rain, or they were collected at UV light, by malaise traps, sifting, or individually. The adults remain for a long time in ant nests, and the digging out of nests is another collecting method used, according to reports from local collectors in Laos and Di Giulio [34]. Some previously recorded species of *Scaphipaussus* were collected from ant nests, mainly of the genus *Pheidole* Westwood, 1839, e.g., *Pheidole ghatica* Forel, 1902; *P. javana* Mayr, 1867; *P. latinota* Roger, 1863; *P*, *pallidula* (Nylander, 1849); *P. plagiaria* Smith, 1860, and *P. wroughtoni*, Forel, 1902 ([1,23,35,36]. Host ants were not obtained for identification.

The type material was studied in the institutions listed below. The recently collected specimens were placed in 50% ethanol for several hours, and then the detached apical part of the abdomen was shortly relaxed in hot 10% KOH to clean the genitalia. Historical specimens from museum collections needed a longer time to soften. They were cleaned and remounted to take photographs with exposed diagnostic characters. The photographs of diagnostic characters were taken using a Canon M6 Mark II digital camera attached to an Olympus SZX16 stereoscopic microscope. Morphometric data from adult males were measured with an ocular grid on an Olympus SZX-16 binocular dissecting microscope. The photo stacks were processed with Helicon Focus, and the final photographs were assembled in figures using Adobe Photoshop software v. 25.2.0. The morphological terms used in descriptions mostly follow Maruyama [23].

The specimen size was measured with a 180-line calibrated scale installed in the eyepiece of an Olympus 16SZX binocular microscope. The following size characteristics are reported: body length from apex of head to apices of elytra (BL); the length of the antennal club, including the posterior basal process if present (ACL); the head width, including eyes (HW), the length of the pronotum along the longitudinal midline (PL), the width of the pronotum taken in the broadest part (PW). The spans designate measurement of maxima and minima in the type series; the holotype data are separately noted. The ratio of the length of the antenna club to its width uses the width in the mid-length of the antennal club.

The material cited in this work is deposited in the museums and collections listed below. The holotypes are deposited in the British Museum, the Musée National in Paris, and in the collection of the first author, which will later be deposited in the National Museum in Prague, Czech Republic.

AKCG—Andreas Kaupp collection, Germany.

BMNH—The Natural History Museum, London, United Kingdom.

FPC—Filip Pavel collection, Vysoká nad Labem, Czech Republic.

IJC—Ivo Jeniš collection, Lutín, Czech Republic.

IMC—Ivo Martinů collection, Olomouc, Czech Republic.

MBC—Michal Bednařík collection, Olomouc, Czech Republic.

MNHN—Musée National d’Histoire Naturelle, Paris, France.

ZFMK—Zoologisches Forschungsinstitut und Museum Alexander König, Bonn, Germany.

## 3. Results


**Taxonomy**
Family Carabidae Latreille, 1802Subfamily Paussinae Latreille, 1806Tribe Paussini Latreille, 1806Subtribe Paussina Latreille, 1807Genus *Paussus* Linné, 1775Subgenus *Scaphipaussus* Fowler 1912: 472 [35]Type species *Paussus boysii* Westwood, 1845. Note. *P. waterhousei* Westwood, 1874 was considered the type species by Maruyama [23].*=Cochliopaussus* Kolbe, 1927Type species *Paussus turcicus* Frivaldszky, 1835*=Semipaussus* Wasmann, 1920Type species *Paussus kannegieteri* Wasmann, 1896

### 3.1. The P. hystrix Group Sensu Maruyama [23]

Type species *P. hystrix* Westwood, 1850.

Diagnosis. Body small- to medium-sized compared to other *Paussus* species, its length 3.2–7.2 mm. Body coloration red to brown, sometimes dark colored or bicolored. The cranium with apparent frontal crest, which is composed of a pair of crescent-shaped, vertical shields. Antennae with fused antennomeres 3–11, antennal club variable in shape, with posterior margin bearing excavation delimited by dorsal and ventral edges. Pronotum strongly divided into anterior and posterior parts by apparent transverse cleft; anterior part the pronotum convex, shaped as transverse cleft medially depressed or excavate with lateral trichomes, posterior part depressed around transverse cleft. Elytra usually broader posteriorly, elytral disc often with pair of glabrous spots, ‘mirror markings’ [23], commonly with rows of bristles. Legs more or less compressed.

Remark. The morphological traits assign the new species described below in the subgenus *Scaphipaussus* Fowler 1912 as delimited by Robertson and Moore [3]. Maruyama [23] defined the *hystrix* group in the genus *Paussus* Linnaeus, 1775 based on the presence of a characteristic frontal crest in the head. The *P. hystrix* group is a part of *Scaphipaussus* now. Newly described five new species are added to the nineteen species listed in the *P. hystrix* group by Maruyama [23]. The *P. hystrix* group now contains six species originally described in *Paussus*, subgenus *Scaphipaussus* (*P*. *drescheri* Reichensperger, 1935, *P. formosus* Wasmann, 1912, *P. hystrix* Westwood, 1850, *P. kolbei* Reichensperger, 1925, *P. tristis* Wasmann, 1912 and *P. waterhousei* Westwood, 1874) and additional two species recently described species in *Paussus* (*P. serraticornis* Nagel et Bednařík, 2013 and *P. zhouchaoi* Wang, 1917). Additional species were placed in the group by Maruyama [23] when the group was established.

The *P. hystrix* group occurs in Southeast Asia and China and contains presumably closely related species. We consider this informal species group helpful for an easy orientation in the diversity of *Paussus*. The *P. hystrix* group belongs to the subgenus *Scaphipaussus* as shared characters support the monophyly of the subgenus.


***Paussus* (*Scaphipaussus*) *fencli* sp. nov.**
(Figure 4A–H)urn:lsid:zoobank.org:act:08A0638A-1618-4642-AC1B-8FD6C19D9CB3

Type material. Holotype, male: China, Guangdong Prov., Datian Ding Mt., 1200–1600 m, 22°16′ N 111°13′ E, 6–7 May 2002, R. Fencl leg. (MBC). Paratypes: female, Kouy–Tchéou [=Guizhou prov., China], R. P. J. R. Chaffanjon, 1903/Museum Paris; female, Kouy-Tchéou, Kouy-Yang [=Guiyang], P. Cavalerie et Fortunat, 1906; male, female, Kouy-Tchéou, Rég. De Pin-Fa, Piére Cavalerie, 1909; male, female, Kouy-Tchéou, P. Cavalerie, 1910 (all MNHN); male, Kan-tschou, Kiang–si, China/coll. Reichensperger; female, Kwelchou, China, Plason/coll. Reichensperger (both ZFMK).

Note. The specimens deposited in MNHN were pinned in the box with a single specimen labeled as *P. sinicus* Westwood, 1850. The specimens deposited in the Museum A. Koenig in Bonn were identified as *P. waterhousei* Westwood, 1874.

Differential diagnosis. The shape of the antennae and pronotum indicates putative relationships of *P. fencli* with the species placed in the *P. hystrix* group in the subgenus *Scaphipaussus*. This species is similar to *P. zhouchaoi* Wang, 2017, which occurs further north. The latter differs in a narrower head and the presence of lateral bristles on the elytra. The new species additionally differs from all relatives in the species group by a slenderer and longer antennal club and has a characteristically robust body with a smooth surface.

Description. Male. Body length 5.8–6.4 mm, concolor reddish brown, various parts of body and appendages dark reddish brown, apex of elytra pale, mouthparts dark brown; most body parts slightly shining with scattered short setae, on lateral margins of elytra and pygidium with long bristles; mirror markings on elytra absent (Figure 4A,G). Head wider than long, narrower than pronotum, with shallow longitudinal median groove reaching middle of frontal crest and symmetrical depressions laterally of it; frontal crest large, massive, with pair of crescent-shaped vertical shields, slightly emarginate at apices in lateral view; enclosed median area of frontal crest with pair of small tubercles; surface in upper part shiny, roughly punctate to rugose; enclosed median depression glabrous and glossy; temples convex, narrower posteriorly; cranium with sparse recumbent setae along eyes and in frons, setae denser around the temples; eyes moderate in size, prominent; neck weakly constricted (Figure 4A,F,G). Mouthparts with transverse, rectangular labrum; maxillary palpomere 2 large, broad and compressed, almost as wide as long; ligula large and broad, broadly rounded at middle. Antenna with scape longer than wide, cylindrical basally, dilated apically; surface rugose, with short setae; fused antennal club flat, elongate, about five times as long as wide with characteristic shape, triangular in cross-section, with a longitudinal depression dorsally; widest at basal margin, subsequently tapering in basal third, widened slightly towards rounded apex; anterior margin of antennal club concave basally, weakly and irregularly wavy; dorsal margin of very narrow excavation with five shallow denticles, each without trichome-bearing hole; ventral margin of excavation laterally straight, only slightly wavy; posterior basal process prominent, but not forming spur; posterior excavation with five shallow circular pits; surface rugose to punctate except for shiny area along denticles (Figure 4A–G). Pronotum transverse, width 1.3 times length, wider than head; frontal part wider than posterior one, raised and slightly depressed medially, laterally acute; surface slightly indented with suberect setae; transverse cleft wide and deep, surface glabrous and smooth; pronotal trichomes large and dense, yellow-orange with dark-colored trichomes in middle; trichome cavity oblique, posterior part with slightly convex lateral margins, medially with wide, longitudinal groove towards scutellum; surface slightly indented with rather short setae (Figure 4A,F,G). Elytra broad, widest apically, strongly convex; humeri rounded; surface sparsely punctate, smooth, sparsely covered with short yellowish, thin setae; posterior third of elytral margin with 6–8 long red-brown bristles; elytral surface without mirror markings. Pygidium with long bristles along margins; disc sparsely pubescent, pygidial setae thinner and less arcuate than setae on elytra.

(Figure 4A,G). Legs robust, femora and tibiae slightly compressed, dilated apically; tibiae almost straight; hind-tibiae broader than mid-tibiae; tarsi with tarsomeres 1–4 unequal in length, with apical margins entire, dorsally straight or inconspicuously emarginate; surface of legs almost smooth with long, suberect setae; claws simple. Male genitalia with continually curved phallus, its apex distinctly emarginate (Figure 4H).

Sexual dimorphism weak, without distinct differences between sexes; females with more robust antennal club than males (Figure 4B–E).

Measurements. BL 5.8–6.4 mm, ACL 1.98–2.29 mm, HW 1.14–1.25 mm, PL 1.20–1.42 mm, PW 1.51–1.61. Holotype, BL 6.1 mm, ACL 2.26 mm, HW 1.21 mm, PL 1.20 mm, PW 1.55 mm, phallus length 1.8 mm.

Distribution. China, Guangdong, Guangxi, and Guizhou provinces. The holotype was collected on the southern slope of the Datian Ding mountain.

Etymology. The species is named in honor of Dr. Rudolf Fencl (Praha, Czech Republic, currently living in the Dominican Republic), the collector of the holotype.


***Paussus* (*Scaphipaussus*) *mawdsleyi* sp. nov.**
(Figure 5A–H)urn:lsid:zoobank.org:act:EE973859-0F6A-4CA0-BA94-E7335C00D7DB

Type material. Holotype, male: Borneo, Brunei, Ulu Temburong NP, Kuala Belalong Field Studies Centre, 4°34′ N 115°7′ E 10 February 2013, malaise trap, J. Ševčík leg. (MBC). Paratypes: female, same data as holotype, but 12 January 2014, sifting, I. Tuf leg. (MBC); 2 females, Borneo, Brunei, 4°34′ N 115°7′ E, April–May 1991, Kuala Belalong FSC, Malaise trap/N. Mawdsley, BMNH1991-173 (BMNH); female, Borneo, Brunei, 4°34′ N 115°7′ E, 8 February 1992, Kuala Belalong FSC, Ground Malaise 9B, 260 m alt., N. Mawdsley NM296 (BMNH).

Differential diagnosis. This species resembles *P. roslii* Maruyama, 2016 but has a slightly different shape of the antennal club. The club is longer, robust, and with a projecting posterior basal process. The pronotum has the posterior part with a wide groove in the middle. Two similar species, *P. mawdsleyi* sp. nov. from Borneo and *P. roslii* Maruyama, 2016 from the Malay Peninsula, differ from the relatives in the *P. hystrix* group in the robust and thick antennal club (Figure 5B,C).

Description. Male. Body length 5.3–6.0 mm, almost uniformly reddish brown, anterior part of pronotum and disc of elytra mostly light colored, margins and posterior parts of body dark brown; lateral and posterior part of elytra with long bristles; mirror markings present (Figure 5A,E). Head with large eyes, strongly transverse, as wide as pronotum, with shallow longitudinal median groove exceeding posterior part of frontal crest; frontal ridge weakly developed; eyes convex, protruding; temples gently convex, less prominent than eyes; occipital suture distinct; frontal crest middle-sized, with a pair of crescent-shaped, quite prominent conical shields; enclosed median area in frontal crest with a pair of strongly convex tubercles which occupy most part of inside of each shield; surface mat, finely, densely punctate with thin setae (Figure 5A,F,G). Mouthparts with maxillary palpomere 2 robust, broad, and compressed. Antennae with scape cylindrical basally, dilated apically; surface rugose, with short setae; antennal club thick, elongate, in male, 3 times and in females about 2,5 times as long as wide, triangular in cross-section, widest basally; flagellar base with a shallow dorsal notch; anterior margin of antennal club slightly concave in basal third; fore edge almost straight; apex rounded; dorsal margin of excavation with five shallow denticles with traces of glands without trichome-bearing openings; ventral margin of excavation almost straight, only slightly undulate; posterior basal process arcuately protruding; posterior excavation deep with five elliptical pits; surface punctate in male, but shining in females, very finely punctate, sparsely covered with short setae in both sexes (Figure 5A–C,E,G,H). Pronotum 1.2 times wider than long, as wide as head, slightly wider than head in females; frontal part wider than posterior one, raised and slightly depressed medially, laterally acute; surface slightly shining except densely punctate lateral and anterior margins, covered with short, suberect setae; transverse cleft wider, deep, surface glabrous and smooth; pronotal trichome large yellow colored; trichome cavity widely opened, almost horizontal laterally, posterior part with slightly convex lateral margins, medially with wide, longitudinal shallow groove towards scutellum; surface matte, densely punctate, densely covered with short setae, setae except transverse cleft (Figure 5A,F,G). Elytra almost parallel-sided, only very slightly widened posteriorly, humeri well-developed, surface densely punctate, matte, uniformly covered with short, recumbent setae; lateral bristles long, forming bundles; mirror markings present. Pygidium not bordered apically; surface glabrous but sparsely covered with setae on disc; marginal pygidial bristles indistinct. Hind wings fully developed (Figure 5A,D,E). Legs relatively strong, femora and tibiae compressed, tibiae dilated apically, very slightly curved; mid and hind tibiae with outer apical process; tarsi with tarsomeres 1–4 unequal in length; surface sparsely covered with long setae; claws simple (Figure 5A,D,E).

Sexual dimorphism conspicuous. The sexes differ in the shape of the antennae; females have a shorter body (Figure 5B,C,G,H).

Measurements. BL 5.3–6.0 mm, ACL 1.41–1.62 mm, HW 1.31–1.36 mm, PL 1.06–1.18 mm, PW 1.26–1.42 mm. Holotype, BL 5.3 mm, ACL 1.62 mm, HW 1.31 mm, PL 1.06 mm, PW 1.30 mm.

Distribution. Borneo (Brunei).

Biology. Four specimens, including the holotype, were collected by Malaise traps, and one specimen was sifted from organic debris. The host ant remains unknown.

Etymology. The specific epithet is a patronym in honor of Norman Mawdsley (United Kingdom), the first collector of the new species.


***Paussus* (*Scaphipaussus*) *bakeri* sp. nov.**
(Figure 6A–E)urn:lsid:zoobank.org:act:7CC1F712-5849-4278-8B0A-0BF3A63CC1E8

Type material. Holotype, male: Philippines, Luzon, Benguet, Baguio, C. F. Baker/Brit. Mus., 1924–486 (BMNH). Note. The specimen was pinned in a small box along with a *Paussus* specimen identified as *P. suavis* Wasmann, 1894.

Differential diagnosis. This species resembles *P*. (*S*.) *kolbei* Reichensperger, 1925 from the Philippines in the shape of pronotum and head. The flattened antennal club has a unique drop-shaped apex of the postero-lateral process. The species has a darker-colored head and pronotum compared to the elytra and long golden pubescence of elytra.

Description. Male. Body length 5.2 mm, head, pronotum and legs dark brown, elytra lighter brown with long bristles; mirror markings present. Head equal in length and width, with shallow longitudinal median furrow not surpassing frontal crest; frontal ridge well developed, more raised; eyes small and flat; short temples; frontal crest flat, less conspicuous than in other species, with pair of slightly curved, flat shields; enclosed median area of frontal crest with three irregular shallow depressions, remaining area wrinkled, cranial surface dull, rugose with inconspicuous setae (Figure 6A,D). Antenna with cylindrical scape; surface rugose, with very short setae; antennal club thick, elongate, three times longer than wide, compressed; widest at basal margin, tapering to apex; basal part marked by shallow dorsal notch; anterior margin almost straight with inconspicuous anterior basal process; apex rounded; dorsal margins of narrow excavation with five low denticles with traces of glands; ventral margin of excavation slightly undulate; posterior basal process characteristically drop-shaped; surface wrinkled, covered very short setae (Figure 6A–D). Pronotum width 0.95 times length, slender than head; frontal part narrower than posterior one, laterally at same level as posterior part, slightly depressed medially; surface rugose; transverse cleft narrow with smooth surface, yellow-orange pronotal trichomes of medium size; trichome cavity opened, slightly oblique laterally; posterior part with slightly convex lateral margins, medially with strait shallow groove; surface matte, densely rugose, almost glabrous (Figure 6A,D). Elytra almost parallel-sided, only very slightly widened in posterior part, gently compressed laterally, humeri weakly developed; surface densely punctate, matte, uniformly covered with short, recumbent white setae; lateral, mesal, and apical parts with very long bristles, yellow-golden colored; each elytron with elongate and relatively dull mirror markings. Pygidium bordered apically; surface covered with short setae; pygidial bristles distinct. Hind wings fully developed (Figure 6A,D). Legs long and thin, femora and tibiae slightly compressed, tibiae dilated apically and very slightly curved convexly outwards; surface sparsely covered with long bristles (Figure 6A,D). Male genitalia with almost regularly curved, subtle phallus (Figure 6E).

Female unknown.

Measurements. Holotype, BL 5.2 mm, ACL 1.48 mm, HW 1.10 mm, PL 1.09 mm, PW 1.05 mm, phallus length 1.6 mm.

Distribution. The Philippines, Luzon island.

Etymology. The species is dedicated to Charles Fuller Baker (1872–1927), the famous explorer of the Philippine fauna and flora and the collector of the unique specimen.


***Paussus* (*Scaphipaussus*) *jendeki* sp. nov.**
(Figure 7A–E)urn:lsid:zoobank.org:act:8E0FCA1C-A88A-4B96-B4FC-4D2541F0A238

Type material. Holotype, male: Laos (central), Khammouan prov., route No. 8, Nakai env., 17°42.8′ N 105°08.9′ E, alt. 560 ± 20 m, 4–5 May 1998, E. Jendek & O. Šauša leg. (MBC).

Differential diagnosis. *Paussus jendeki* sp. nov. resembles *P. hystrix* Westwood, 1849 from which it differs in narrower and flattened antennal club. Slender tibiae resemble those of *P. jengi* Maruyama, 2016. The new species has lightly colored elytra that are narrower and longer than those of *P. hystrix*. Additionally, the new species is characterized by a rough surface between denticles (i.e., raised notches) at the posterior dorsal margin of the antennal club (Figure 7A–D). This area is smooth in *P. hystrix* (Figure 7H). There is sexual dimorphism in the antennal club of *P. hystrix* when females have regularly smoother surfaces than conspecific males. The female of *P. jendeki* is unknown.

Description. Male. Body length 5.9 mm, head, pronotum and elytra light brown, but antennae, legs, and ventral side dark reddish brown, elytra with very long bristles; mirror markings present (Figure 7A,D). Head wider than long, with shallow, short median furrow reaching anterior part of frontal crest, its sides weakly convex anteriad; frontal ridge weakly developed; eyes rather large, slightly prominent; temples short; frontal crest large, with a pair of crescent-shaped, vertical shields, gently convex dorsad, their apices grooved, and slightly sinuate in lateral view; enclosed median area of frontal crest with a pair of flat tubercles; surface matte, finely, densely punctate; frontal crest and enclosed median area glabrous and glossy; sparsely covered with minute setae; neck gently constricted, narrower than frontal part of cranium (Figure 7A,E). Antenna with scape cylindrical basally, dilated apically; surface rugose with long suberect setae; antennal club flattened, elongated, three times longer than wide, compressed; widest at third of club, nearly parallel-sided; basal portion marked by shallow dorsal notch; anterior margin of antennal club almost straight basally; apex rounded; dorsal margin of excavation with five low denticles with traces of glands; ventral margin corrugated; posterior basal process small, not projecting but pointed; posterior excavation very narrow (twice slenderer than *P. hystrix*, Figure 7D), deep with five elliptical pits; surface wrinkled, rugose, including areas between denticles, covered with very short setae (Figure 7A–C,E). Pronotum 1.15 times wider than long, wider than head; anterior part wider than posterior one, slightly depressed medially, laterally acute in dorsal view; surface rugose, covered sparsely with long suberect setae; transverse cleft somewhat wider, deep, its surface smooth; pronotal trichome small, red-colored hair; trichome cavity widely opened, almost horizontal laterally, posterior part with slightly convex lateral margins; surface slightly rugose, covered with long setae except transverse cleft (Figure 7A,E). Elytra nearly parallel-sided, slightly convex, widest in middle, relatively narrow and long, gently compressed laterally, humeri weakly developed; surface densely punctate, matte, uniformly covered with long, erect, white setae; lateral, mesal, and apical red-haired bristles very long, forming longitudinal rows; mirror markings present. Pygidium bordered apically; surface finely and densely dotted; pygidial bristles distinct, long. Hind wings fully developed (Figure 7A,E). Legs long and thin, femora and tibiae slightly compressed, tibiae very slightly dilated apically; tarsi with tarsomeres 1–4 unequal in length; surface sparsely covered with longer thin setae (Figure 7A,E).

Female unknown.

Measurements. Holotype, BL 5.9 mm, ACL 1.8 mm, HW 1.22 mm, PL 1.10 mm, PW 1.33 mm, maximum width of the antennal club 0.68 mm, elytral length 3.9 mm, width in the middle part of elytra 2.34 mm, maximum width of hind tibia 0.22 mm, ditto *P. hystrix* 0.27–0.38 mm; maximum width of middle tibia 0.18 mm (ditto *P. hystrix*: 0.24–0.30 mm).

Distribution. Central Laos.

Biology. The host ant remains unknown. The unique specimen was collected by a UV light trap installed in the place shown in Figure 3A.

Etymology. The specific name is derived from the personal name ‘Jendek’, and the species is dedicated to Eduard Jendek, one of the collectors of the holotype.


***Paussus* (*Scaphipaussus*) *saueri* sp. nov.**
(Figure 8A–E)urn:lsid:zoobank.org:act:75E9B042-B025-4DC9-8FE1-0CC7041A069E

Type material. Holotype, male: India, Nilgiri hills, Kunchapanai, 77°0′ E 11°20′ N, 12–22 May 1994, R. Sauer leg. (MBC).

Differential diagnosis. This species is similar to *P. hystrix* Westwood, 1849 from which it differs in a shorter and broader antennal club, the coloration of the elytra, slender anterior half of the pronotum, wide metathoracic tibiae and different pubescence on elytra (Figure 8).

Description. Male. Body length 5.1 mm, light red-brown colored, dark brown to black in various parts of body and appendages; head, pronotum, antenna, anterior part of elytra and legs reddish brown; apex of elytra yellowish; pronotum laterally, posterior parts of elytra, mirror markings and femora basally dark brown to black; most body parts rugose punctate (Figure 8A,D). Head wider than long, with short median furrow reaching to anterior part of frontal crest, with vertex convex; frontal ridge curved; eyes small, flat; temples short; frontal crest small, with pair of strongly curved shields, their apices grooved; opened median area in frontal crest with glossy pits without tubercles; surface rugose without enclosed median areas of crest; minute setae imperceptible; neck gently constricted, narrower than frontal area of cranium (Figure 8A,D). Antenna with scape cylindrical basally, widest in middle; surface rugose with short suberect setae; antennal club flattened, 2.5 times longer than wide, widest at basal third, triangular in cross-section, tapering to rounded apex; basal portion marked by dorsal notch; anterior margin of antennal club slightly basally concave, almost straight; dorsal margin of excavation with five low denticles with traces of glands; ventral margin corrugated, convex; posterior basal process small, not projecting, obliquely cut; posterior excavation with five elliptical pits; surface slightly rugose, but shining, sparsely covered with very short setae (Figure 8A–C,E). Pronotum almost as wide as long, wide as head; frontal part slightly narrower than posterior one, with rounded sides and depressed medially; surface rugose, with relatively thicker suberect setae; transverse cleft somewhat wider, deep, its surface smooth; yellow-orange pronotal trichomes of medium size; trichome cavity opened, oblique laterally, posterior part with slightly convex lateral margins, medially with deeper groove; surface rugose, covered with long setae except transverse cleft (Figure 8A,D). Elytra almost parallel-sided, only very slightly widened in posterior part, gently compressed laterally; humeri weakly developed; surface sparsely but deeply punctate, smooth to shiny, with short and whitish setae; mesal and apical parts with scattered long setae, long, dark-orange bristles on lateral margins, arranged in bundles in longitudinal row; mirror markings smooth. Pygidium bordered apically; surface finely dotted; pygidial bristles distinct, long. Hind wings fully developed (Figure 8A,D). Legs strong, robust, femora broad, compressed, tibiae flattened, hind tibiae convex on outer margin, widened to apex; tarsi with tarsomeres 1–4 equal in length; surface sparsely covered with thin setae. Male genitalia with an irregularly curved phallus (Figure 8E).

Female unknown.

Measurements. Holotype, BL 5.1 mm, ACL 1.26 mm, HW 1.08 mm, PL 1.08 mm, PW 1.06 mm, phallus length 1.5 mm.

Distribution. Southern India, Tamil Nadu state.

Biology. Host ant unknown. The unique specimen was collected at the light.

Etymology. This species is named in honor of Roman Sauer (Praha, Czech Republic), who collected the unique specimen.

### 3.2. The Species Assigned to Scaphipaussus Sensu Robertson & Moore [3]

Some *Scaphipaussus* cannot be assigned to the *P. hystrix* group and differ in the entire crest that is not divided into two separate parts. They have a C-shaped crest or conical protrusion (protuberance by Fowler, 1912). Additionally, they have characteristic shell-shaped antennal clubs with a deep and oval posterior excavation. Mirror markings are usually absent. Similar species were earlier placed in *Cochliopaussus* Kolbe, 1927. These new *Cochliopaussus*-like species include *P. annamensis* sp. nov. and *P. phoupanensis* sp. nov.


***Paussus* (*Scaphipaussus*) *annamensis* sp. nov.**
(Figure 9A–E)urn:lsid:zoobank.org:act:6B4CCECF-14B6-49A5-AAE6-B6BB84FDC85F

Type material. Holotype, male: [Vietnam] Annam, Phuc-Son, Nov. Dez., H. Fruhstorfer/Muséum Paris, 1952, Coll. R. Oberthür (MNHN).

Differential diagnosis. The new species resembles *P. corporaali* Reichensperger, 1927 the general habitus, and *P. atheruri* Luna de Carvalho, 1960 in the presence of bristles, forming bundles on the elytra. The new species is characterized by the unique shape of the antennal club (Figure 9B,C), big eyes, and the size of a C-shaped crest (Figure 9D). The male of *P. annamensis* sp. nov. has a long and slender antennal club compared to *P. corporaali* (Figure 9).

Description. Male. Body length 4.7 mm, almost uniformly reddish brown; posterior margin of antennal excavation, posterior part of pronotum laterally and disc of elytra dark brown, elytra with long bristles forming bundles in longitudinal rows; elytra compressed laterally, mirror markings absent (Figure 9A,D). Head wider than long, with short median furrow reaching to anterior part of frontal crest, its sides depressed; conically tapering to the anterior margin with rounded frontal ridge; vertex with large C-shaped crest, its anterior part open (Figure 9A,D); surface matte, rugose; eyes large and prominent; temples slightly convex, surface sparsely covered with short setae; neck little constricted (Figure 9A,D). Antenna with scape subcylindrical, rugose, with short suberect setae; antennal club 2.3 times longer than wide, widest at middle of club, tapering to rounded apex; basal portion marked by oval dorsal notch; anterior margin of antennal club almost regularly concave, anterior basal angle absent; dorsal margin of excavation wavy with five low denticles; ventral margin slightly sinusoidally curved; posterior basal process projecting to tip; posterior excavation shallow with five pits; surface slightly rugose, but shining, sparsely covered with short and thin setae (Figure 9A–D). Pronotum 1.2 times wider than long, as wide as head; slender posteriorly, slightly depressed medially, laterally acute in dorsal view; transverse cleft narrower, its surface smooth; pronotal trichomes extensive and long, yellow-orange colored; trichome cavity widely opened; posterior part with slightly convex lateral margins; medially with shallow square depression; surface slightly rugose, shiny, pronotum with short setae (Figure 9A,D). Elytra convex, widest in posterior third, gently compressed laterally, without mirror markings; humeri slightly raised; surface finely rugose, shiny, with long, whitish setae; mesal and apical parts with long bristles forming bundles in two longitudinal rows on each elytron. Hind wings fully developed. Pygidium blade-like, bordered apically; surface finely dotted; marginal pygidial setae long, dense, twisted. Legs slender, slightly compressed, surface sparsely covered with thin setae (Figure 9A,D). Male genitalia with irregularly curved robust phallus and with short parameres (Figure 9D).

Female unknown.

Measurements. Holotype, BL 4.7 mm, ACL 1.32 mm, HW 1.01 mm, PL 0.88 mm, PW 0.99 mm, phallus length 1.4 mm

Distribution. Central Vietnam, southwest of Da Nang.

Etymology. The specific epithet refers to the distribution of this species. Annam, shown on the locality label of the holotype, was an earlier designation of the central region of the modern Republic of Vietnam when this species was collected.


***Paussus* (*Scaphipaussus*) *phoupanensis* sp. nov.**
(Figure 10A–E)urn:lsid:zoobank.org:act:3EDD5033-8295-4BA6-94F3-2E7B7A4AD3B7

Type material. Holotype, male: Laos NE, Hua Phan prov., Mt. Phu Pane, 103°59′ E 20°12′ N, ±1200 m, 10 May 2011, collected in ant nests, Lao collectors leg. (MBC). Paratypes: male, Laos NE, Hua Phan prov., Ban Saluei, 104°01′ E 20°12′ N, 1300–1900 m, 7 May 2011 (MBC); two males, two females, same data as the holotype, but 1200–1600 m, 10–22 May 2011, (MBC, a female deposited in IMC); female, Laos, Huaphanne prov., Mt. Phu Pane, 1200–1900 m, Ban Saluei village env., 21–30 Apr. 2017, 103°59′ E 20°12′ N, A. & R. Hergovits and Lao collectors leg. (MBC); male, Vietnam, Lao Cai prov., Sa Pa, 1 Apr. 2018, D. Rydzi leg. (MBC).

Differential diagnosis. The new species resembles in the general appearance some Oriental species, e.g., *P. quadricornis* Wasmann, 1899 and *P. bowringi* Westwood, 1850. *P*. (*S*.) *phoupanensis* sp. nov. differs from these two species in flat, non-protruding temples and very sparse pubescence with very long setae and thin bristles on the elytra. It differs from *P. atheruri* Luna de Carvalho, 1960 in the shape of the antennal club. *P. phoupanensis* is characterized by a long posterior basal process (Figure 10B).

Description. Body robust with length 4.6–5.3 mm, whole body and legs dark reddish brown; posterior margin of antennal excavation, area around pronotal cleft, and disc of elytra almost black; ventral part of head, prothorax, mesosternum, and metasternum slightly darker than dorsum, apex of elytra reddish. Most body parts finely punctate, surface of elytra with sparse very long, thin setae; mirror markings absent (Figure 10A,D). Head wider than long, narrower than pronotum, tapering from eyes to clypeus; median groove absent; clypeus slightly inclined down with frontal ridge of triangular shape; frons raised into conical protrusion (i.e., protuberance) and furnished with rather large cavity; eyes small; temples convex, narrower posteriorly; surface covered with long setae; neck little constricted, wider than clypeus (Figure 10A,D). Antenna with scape strongly punctate, with tenuous erect setae; antennal club twice longer than wide, shell-shaped, expanded apically; sharply keeled on anterior margin and another side deeply and broadly excavate; basal portion broken by elongated longitudinal notch; dorsal margin of posterior excavation sinusoidally curved with five deep furrows, ventral margin obtusely dentate with six teeth, and with thin setae, posterior basal process acute and broadly bent; whole club, except dorsal margin, dull and thickly punctured, excavation glabrous, smooth, and shiny (Figure 10A–D). Pronotum 1.1 times broader than long at midline, with base and apex of about equal breadth; anterior part short, strongly raised, with sides deeply excised and bidentate, so that four prominences are evident on raised basal margin; posterior part deeply and broadly impressed frontally, with transverse cleft, longitudinally channeled; pronotal trichome yellow-orange colored; posterior part with distinctly convex lateral margins; surface slightly rugose, in places shiny, pronotum with sparse and long setae (Figure 9A,D). Elytra slightly convex, widest in posterior third, gently compressed laterally, without mirror markings; humeri weakly raised; surface finely rugose, shiny, with long, whitish setae; lateral parts with long bristles. Hind wings fully developed. Pygidium blade-like, bordered apically; surface finely dotted; marginal pygidial bristles long, dense, and twisted. Legs slender, slightly compressed, mid and hind tibiae weakly curved; surface sparsely covered with thin setae (Figure 10A,D). Male genitalia with irregularly curved apically wide phallus and with long parameres (Figure 10D).

Measurements. BL 4.6–5.3 mm, ACL 1.32–1.49 mm, HW 1.10–1.21 mm, PL 0.96–1.10 mm, PW 1.20–1.36 mm, phallus length (paratype) 1.7 mm. Holotype, BL 5.3 mm, ACL 1.41 mm, HW 1.18 mm, PL 1.16 mm, PW 1.32 mm.

Distribution. Northeastern Laos (Phou Pane mountain), northern Vietnam. Currently, only one record is known from the mountains in the vicinity of Sa Pa in northern Vietnam.

Biology. The specimens from the Phou Pane region (northern Laos) were mostly collected around the village Ban Saluei by local field collaborators. Specimens were collected individually on vegetation and around lights but mainly by digging up ant nests with paussids. Still, the host ant was not attached to any specimen obtained and could not be identified.

Etymology. The species epithets point to the type locality. We use the spelling Phou Pane as listed in most maps.

### 3.3. Additional Species Assigned to Scaphipaussus Sensu Robertson & Moore (2017) [3]

Robertson & Moore [3] listed in the Supplements of their study four species in *Scaphipaussus* (*P. lanxangensis* Nagel, 2009, *P. serraticornis* Nagel & Bednařík, 2013, *P. drumonti* Maruyama, 2014 and *P. masaoi* Maruyama, 2014). These species were earlier assigned to *Paussus* without any subgeneric placement. *Paussus serraticornis* resembles *P. siamensis* Maruyama, 2016 from the *P. hystrix* group and is listed in the *P. hystrix* group now (see Appendix A). *P. masaoi* resembles *P. bilyi* sp. nov. described below. *Scaphipaussus* not included in the *P. hystrix* group share principal diagnostic characters of the subgenus and differ by the absence of characters defining the *P. hystrix* group. For example, these species differ in the shape of the antennal club. Following the earlier placement of some similar species [3], we place *P. bilyi* sp. nov. and *P. haucki* sp. nov. in *Scaphipaussus*.


***Paussus* (*Scaphipaussus*) *bilyi* sp. nov.**
(Figure 11A–H)urn:lsid:zoobank.org:act:EF4FBF87-2C14-4FDA-85B0-E7479E908D9F

Type material. Holotype. Male: Thailand, Chiang Mai prov., Ban San Pakia, 98°49′ E 19°19.4′ N, ±1400 m, 11–19 May 1998, M. Bednařík leg. (MBC). Paratypes: 2 males, Thailand, Chiang Mai prov., Ban San Pakia, 1700 m, 25 April–7 May 1996, S. Bílý leg. (MBC, IJC); five males, two females, same data as the holotype (MBC, one specimen deposited in AKCG); one spec., same data as the holotype, but R. Veigler leg. (IJC); male, same data as the holotype, but 11–15 May 1998, V. Kubáň leg. (MBC); male, Thailand N, 100 km NE of Nan, Doi Phu Kha N. P., 20–25 April 2004, F. Pavel leg. (FPC).

Differential diagnosis. This species is very similar to *P. masaoi* Maruyama, 2014, but differs in the rugose surface of the antennal club in both sexes. (Figure 11). The two species, which occur together in northwestern Thailand, differ from other *Scaphipaussus* in the unique shape of the antennae.

Description. Male. Body slender, length 4.6–5.2 mm, almost black in color; head, antennae, mouthparts, anterior part of pronotum, elytral suture, tibiae and tarsi reddish brown; dorsal surface of head, antennal clubs, pronotum, legs and pygidium smooth, shining, but elytra sparsely punctured; all parts of body covered with short, sparse pubescence with exception of mostly glabrous head, pronotum, and pygidium; elytra without mirror markings (Figure 11A,D). Head at frontal part with shallow, wide, longitudinal groove, its sides depressed; vertex with large, C-shaped crest with anterior margin open, outlined by shallow, thin groove, and its opening connected to groove on frontal part; surface finely, sparsely punctured, almost glabrous, but along eyes with sparse, short setae; eyes large, prominent; temples distinctly convex, surface sparsely covered with short setae; neck strongly constricted (Figure 11A,D,F,G). Mouthparts with transverse labrum; maxillary and labial palpi cover mouth opening from beneath; maxillary palpomere 2 large, broad and compressed; ligula broadly rounded at middle. Antennal club saber–curved and strongly compressed, slightly convex with rugosely punctate surface, sparsely covered with short setae; posterior margin simple, with sharp basal projection (Figure 11A–D,G,H). Pronotum strongly divided by longitudinal transverse furrow with trichomes; anterior part wider than posterior one, with four protruding lobes dorso-laterally; posterior part with parallel, slightly concave lateral margins; medially with deep, wide, longitudinal furrow; surface of anterior part sparsely covered with minute setae, posterior part glabrous (Figure 11A,D,G). Elytra somewhat expanded, widest around posterior third; surface moderately uniformly punctured, covered with short, thin setae, but along apical margin glabrous (Figure 11A,D). Pygidium with glabrous and shining disc, with circular dense bristles. Legs long; tibiae slightly compressed with hind tibiae broader than mid and fore ones; outer apical angle of hind tibiae obliquely cut; tarsi with segments of unequal length and ventral pubescence. Male genitalia as in Figure 11E.

Females similar to males, but with subtle antennae and large crest.

Measurements. BL 4.6–5.2 mm, ACL 1.78–1.96 mm, HW 0.97–1.03 mm, PL 0.80–0.93 mm, PW 1.05–1.15 mm. Holotype, BL 5.0 mm, ACL 1.90 mm, HW 1.03 mm, PL 0.92 mm, PW 1.10 mm, phallus length 1.45 mm.

Distribution. Northwestern Thailand.

Biology. The specimens collected in the vicinity of Ban San Pakia were swept from the vegetation on a burnt field at midday during the drizzle; one specimen sat on the leaves of lower vegetation close to the secondary forest, and one specimen was running on a fallen tree (pers. observ. by the first author). *P. lanxangensis* Nagel, 2009 occur sympatrically with the new species. The individual from Doi Phu Kha National Park was also collected during the day after rain sitting on the grass. The nests of potential host ants were noted in nearby places, yet the specific host remains unknown.

Etymology. The species name is a patronym dedicated to Dr. Svatopluk Bílý, the deceased curator of the beetle collection in the National Museum in Prague, Czech Republic, who collected the first specimen of the new species.


***Paussus* (*Scaphipaussus*) *haucki* sp. nov.**
(Figure 12A–F)urn:lsid:zoobank.org:act:F70E6CF4-0BBA-4A23-8FD1-E5C5771AECD3

Type material. Holotype, female: Thailand NE, Loei prov., Phu Ruea N. P., 17°30′ N 101°21′ E, 1100 m, 6–9 April 1999, D. Hauck leg. (MBC).

Differential diagnosis. The shape of the antennal club is unique within the *Paussus* (Figure 12B–D,F). *Paussus haucki* resembles *P. lanxangensis* Nagel, 2009 in the habitus and, in some aspects, also *P. drumonti* Maruyama, 2014. The new species differs in the compact elliptical antennal club with two basal projections, one on the posterior margin and the second ventrally. The species belongs to a large group of species characterized by a deep transverse pronotal furrow, but with variable shapes of the frontal protrusions.

Description. Male unknown. Female. Body relatively robust, 4.2 mm long, almost dark brown- to black colored; head, dorsal part of pronotum, antenna, anterior part and apex of elytra, femora apically, tibiae, and tarsi reddish brown. Most body parts finely punctate, head surface rough; all body parts covered with long, mostly sparse pubescence except glabrous part around pronotal transverse cleft and pygidium; mirror markings absent (Figure 12A,E). Head wider than long, narrower than pronotum, tapering from eyes to clypeus; with depression frontally, sides depressed; frontal ridge well developed frontally; vertex with unique shaped crest with pair of curved and very rugose vertical shields; enclosed median area with sharp longitudinal ridge; remaining area wrinkled; eyes small; temples convex, protruded; dull, rugose, with long setae; neck strongly constricted (Figure 12A,D,F). Antenna with long and stout scape, rounded to triangular in cross-section, tapering apically, strongly punctate with erect setae; antennal club 1.8 times longer than wide; antennal club shape complex, margins with apparent protuberances; compact elliptical club with hump on dorsal side and with slender posterolateral cavity and two posterior basal processes, one projecting posteriorly and other downward at angle of approximately 90°; surface slightly sparsely punctate and shiny, covered with long setae (Figure 12B–D,F). Pronotum 1.1 times broader than long, strongly divided into anterior and posterior part by deep transverse furrow; anterior part laterally rounded; divided by longitudinal groove and thus forming two lobes slantingly projecting backwards; transverse cleft with small trichomes, light grey to brown colored; posterior part with parallel, slightly concave lateral margins; medially with deep groove, separating two round, lustrous tubercles, longitudinally diagonally channeled on sides; surface of anterior part sparsely covered with long setae, posterior part almost glabrous (Figure 12A,D,E). Elytra almost parallel-sided, widest in second third of their length; surface uniformly densely punctured, covered with long, thin, and erect setae (Figure 12A,E). Pygidium with glabrous and shining disc, apically with blade-like, semicircular margin, bent medially, with thin, short setae. Legs stout; tibiae slightly compressed and curved, all tibiae broader apically; tarsi with tarsomeres 1–4 subequal in length; ventrally setose (Figure 12A,E).

Measurements. Holotype, BL 4.2 mm, ACL 1.03 mm, HW 0.87 mm, PL 0.86 mm, PW 1.02 mm.

Distribution. Northern Thailand; the type locality only.

Biology. Only a single specimen is known from the type locality. It was swept off the vegetation in a meadow in an open part of the forest. Therefore, the host ant remains unknown.

Etymology. The species name ‘*haucki*’ is a patronym dedicated to David Hauck (Brno, Czech Republic), who collected the unique specimen of this species.

### 3.4. Further Examined Material


***Paussus* (*Scaphipaussus*) *madurensis* Wasmann, 1913**
Paussus madurensis Wasmann, 1913: 381 [37].(Figure 13A–E)

Examined material. Male, S. India, Tamil Nadu state, Palni Hills, 30 km E of Munnar, Top Station, 1900 m, May 1994, Z. Kejval & R. Sauer leg. (MBC); female: [India] Chambaganor, Madura, Inde, Guy Babault, 1913, Museum Paris, Coll. Guy Babault, 1930 (MNHN).

Remarks. There is little information available on *P. madurensis*, and we identified two specimens in the historical collections and newly collected material form Southern India. Here, we illustrate the female deposited in the collection of the Natural History Museum in Paris. According to Wasmann’s original description, the studied specimens slightly differ in the coloration of the head, antennae, and tarsi. The female has a more prominent, wrinkled frontal crest, a wide pronotum, and slightly more robust legs with wide tibiae of mesothoracic legs (Figure 13A–E).

Measurements. Female, BL 5.5 mm, ACL 1.50 mm, HW 1.27 mm, PL 1.25 mm, PW 1.38 mm.

Distribution. Southern India, Tamil Nadu state.

Biology. The specimen from the Top Station locality was active on the forest ground during the day.


***Paussus* (*Scaphipaussus*) *corporaali* Reichensperger, 1927**
Paussus corporaali Reichensperger, 1927: 303 [38](Figure 14A–D)

Examined material. Female: Java occident., Sukabumi, 2000′, 1893, H. Fruhstorfer (MNHN).

Measurements. Measurements. Female, BL 4.8 mm, ACL 1.01 mm, HW 1.03 mm, PL 0.86 mm, PW 0.94, width in the middle of elytra 1.82 mm.

Remark. We found a specimen in the collection of the Natural History Museum in Paris which we identified as *Paussus corporaali* Reichensperger, 1927. As the original description is accompanied only by a low-quality photograph, we present here illustrations of the additional specimen. It slightly differs in missing bristles along the lateral edges of the elytra, a smaller crest in the frons, and a somehow shorter and rounded antennal club (Figure 14). We have not studied the holotype of *P. corporaali* and do not consider these differences as significant enough to propose a new species at this moment.

## 4. Discussion

The incongruence between morphological and molecular delimitation of the *Paussus* subgenera documented by Robertson and Moore [3] is substantial. The situation can preferably be solved by denser sampling of future molecular analyses. Yet, the material for DNA isolation is currently unavailable due to the rareness of Asian *Paussus*. Maruyama (2016) noted that a ten- to fourteen-day collecting trip typically yields a single species in a single or a few individuals. Similarly, Robertson and Moore [3] faced problems with sampling and pointed to the fact that some species have not been collected in the last hundred years. We have the same experience, and most species available for this study were randomly collected by colleagues who spent several weeks in the field each year. At least partial improvement of the *Paussus* taxonomy can be reached by investigating the morphology of dry-mounted specimens. Here, we increased the number of species formally placed in the subgenus *Scaphipaussus* by ~10%. With the assignment to the subgenus, we mainly depend on the presence of the frontal crest and distinctive characteristics that define the *P. hystrix* group [23]. Most species earlier placed to *Cochliopaussus* currently belong to the African *Paussus incertae sedis*, yet some are potentially related to *Anapaussus* and *Hylotorus*, widely distributed and highly diverse in Africa. Species potentially related to Afrotropical species earlier assigned to *Cochliopaussus* have not been reported from the eastern part of the Oriental region [3].

The presence of a characteristic crest was noted by earlier authors [29,35,39,40]. Maruyama [23] listed the presence of the crest as one of the diagnostic characters defining the *P. hystrix* group. The whole structure consists of two isolated moon-shaped crests, and it is well-developed in the species from East Asia (Figure 4, Figure 5, Figure 6, Figure 7, Figure 8, Figure 9, Figure 10, Figure 11 and Figure 12). Still, it is much weaker in the Palearctic *P. turcicus* that belongs to *Scaphipaussus* based on the molecular phylogeny but was earlier placed in the subgenus *Cochliopaussus*. The presence/absence of the crest is far from a binary character state, and the gradual loss of this structure presents a challenge. The described species show a variable morphology of the frontal protrusion. Although generally reliable for the definition of *Scaphipaussus*, there are cases when we cannot be sure. The vertex usually has a well-defined frontal crest (*P. fencli* sp. nov., *P. mawdsleyi* sp. nov., *P. bakeri* sp. nov., *P. jendeki* sp. nov. and *P. saueri* sp. nov.) or the C-shaped crest that is open anteriorly (*P. annamensis* sp. nov. and *P. bilyi* sp. nov.). Alternatively, the vertex bears a conical protrusion with a terminal cavity (*P. phoupanensis* sp. nov.) akin to some species from earlier *Cochliopaussus*, or the crest is limited to a flat wrinkled area with a furrow (*P. haucki* sp. nov.). Unlike the problematic situation when *Scaphipaussus* is considered as a whole, the presence of a well-developed crest in the *P. hystrix* group is helpful for easily delimitating a group of closely related East Asian species.

We found extremely high variability in the shape of the antennal club of *Scaphipaussus* species. Many species have prolonged antennal clubs with a row of small denticles along the postero-lateral margin (Figure 4, Figure 5, Figure 6, Figure 7, Figure 8 and Figure 9). The denticles can be inconspicuous (*P. fencli* sp. nov.) with only a slightly convex posterior portion of the club (Figure 4A–E). Such a club type is characterized by a high length/width ratio. The denticles can be better marked, and simultaneously, the convex part of the posterior margin becomes more prominent and can form an apparent cavity and long poster-basal process (*P. phoupanensis* sp. nov., Figure 10B,C). The species with a deep and oval posterior excavation have much shorter and more robust clubs. Although the postero-basal process of these species can be slightly prominent (Figure 5B, Figure 6B and Figure 9B), in one species, *P. phoupanensis*, the process can be quite long (Figure 10B). The presence of a cavity seems to be linked with the presence of the prominent postero-basal process. An additional type of antennal club is represented by the very simple shape of the club, for example, *P. bilyi* sp. nov. (Figure 11B,G) and *P. masaoi* Maruyama, 2014. The antennae play a role in the interaction with host ants in the nest [5,36] and seem to be under intense selective pressure. The diagnostic value can be used at the species-level identification, but it seems to tell a little about the relationships within the subgenus *Scaphipaussus* and the entire genus *Paussus*.

Although we are limited by the number of individuals, we assembled a relatively large series of these rare beetles, and we can consider the intraspecific variability of antennal clubs. The sexual dimorphism is low, and the sexes mostly differ in the robustness of the antennal club. We found an apparent sexual dimorphism in the shape and surface structure of antennal clubs. For example, we noted sexual dimorphism in *P. hystrix*, *P. mawdsleyi*, *P. indosinicus*, and *P. lanxangensis*.

The taxonomy of many beetle groups is based on the morphology of male genitalia, and Maruyama [23] noted a characteristic apex of the phallus in the *P. hystrix* group. We dissected the male genitalia of almost all species (some are only known in females, and some species are known in unique, fragile specimens, which we preferred not to dissect). We found similar phallic apices in all studied species and cannot propose any well-defined trait in the male genitalia for reliable identification. Nevertheless, the photographs of male genitalia are included in the illustration presented in the study whenever possible to provide comprehensive information about species of *Scaphipaussus* and the *P. hystrix* group, respectively.

The unavailability of properly fixed material is a serious obstacle to further progress in the systematics of Asian *Paussus*. There is little chance to rapidly assemble enough material for a meaningful project, but the morphology-based alpha-taxonomy should prepare a framework for further studies. *Paussus* is a fascinating group of myrmecophilous beetles [3,5,8,20,34], and their diversity surely deserves our attention.

## Figures and Tables

**Figure 1 insects-14-00947-f001:**
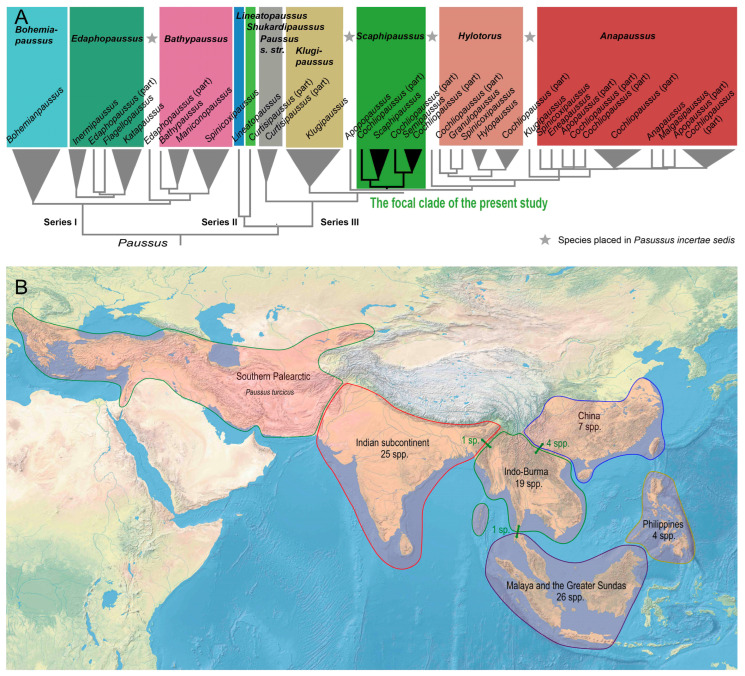
(**A**) The diagram of DNA-based phylogenetic relationships of *Paussus* subgenera modified from Robertson & Moore [3]; the terminal labels designate original placement as reported by the authors in their Figure 18 [3], and the genera listed in the upper part of the figure in the bold font designate the classification proposed in the aforementioned study in Figure 21 [3]. (**B**) Distribution of the earlier described species of the subgenus *Scaphipaussus*, including species transferred from the earlier synonymized subgenera *Cochliopaussus* and *Semipaussus* in this study. The arrows and numbers of species shown in the green color designate the species reported from two neighboring subregions.

**Figure 2 insects-14-00947-f002:**
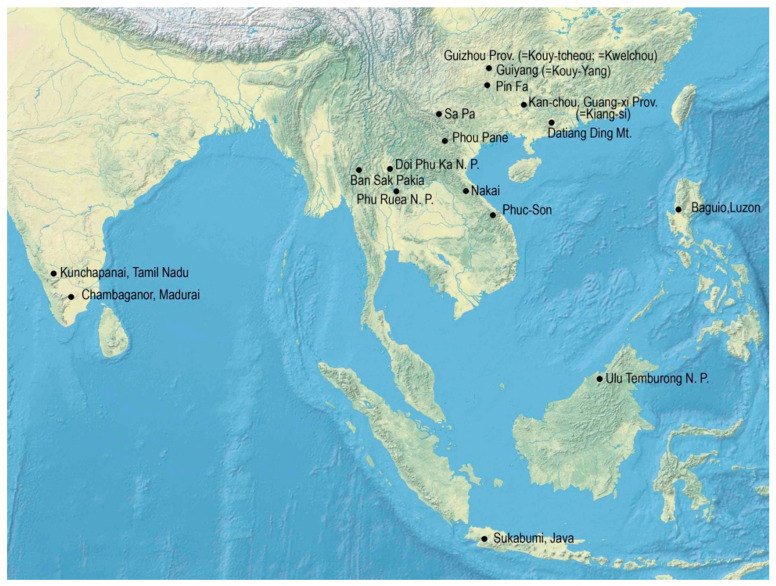
The sampled localities.

**Figure 3 insects-14-00947-f003:**
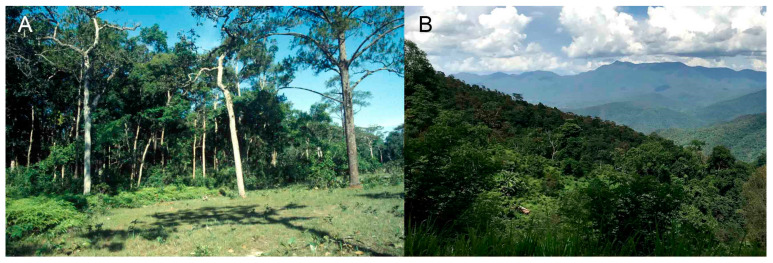
(**A**) Laos, Nakai env., highland plain with sands and pines, habitat of *Paussus jendeki*, type locality (photograph by E. Jendek, 1998). Recently, the habitat was flooded by the dam. (**B**) Thailand, Ban San Pakia, the secondary forest near a village with interspersed fields, the *P. bilyi* type locality (photograph by S. Bílý, 1996). Both photographs are published under open access policy and their authors gave us the permission to publish them.

**Figure 4 insects-14-00947-f004:**
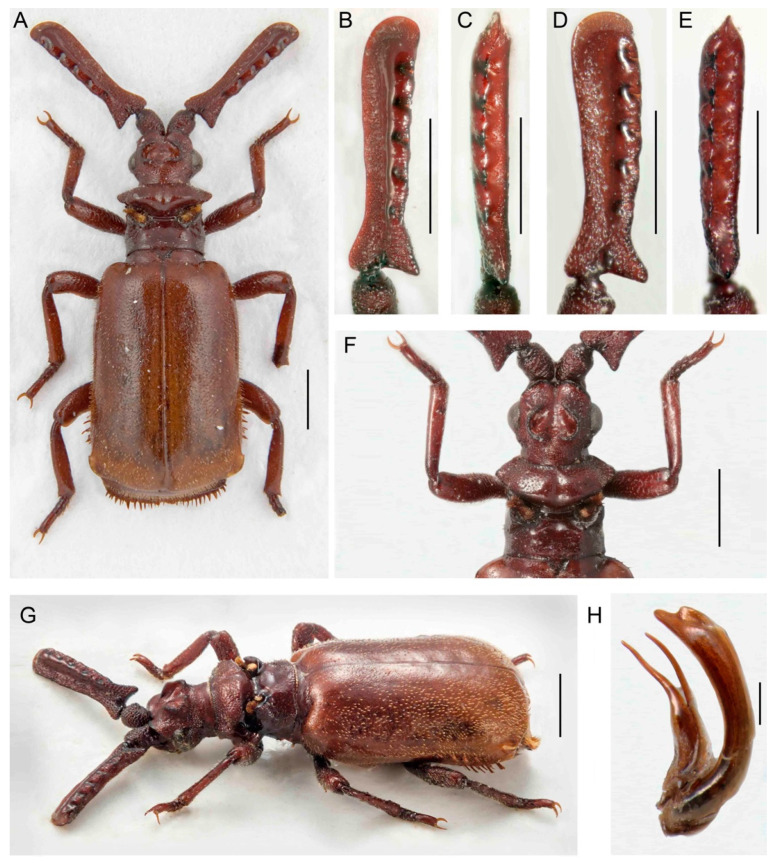
*Paussus fencli* sp. nov., holotype: (**A**) Habitus; (**B**,**C**) antennal club in dorsal and lateral views; (**F**) head and pronotum; (**H**) male genitalia; paratype, female: (**D**,**E**) antennal club in dorsal and lateral views; (**G**) habitus in dorsolateral view. Scales 1 mm (**A**,**F**,**G**), 0.5 mm (**B**–**E**,**H**).

**Figure 5 insects-14-00947-f005:**
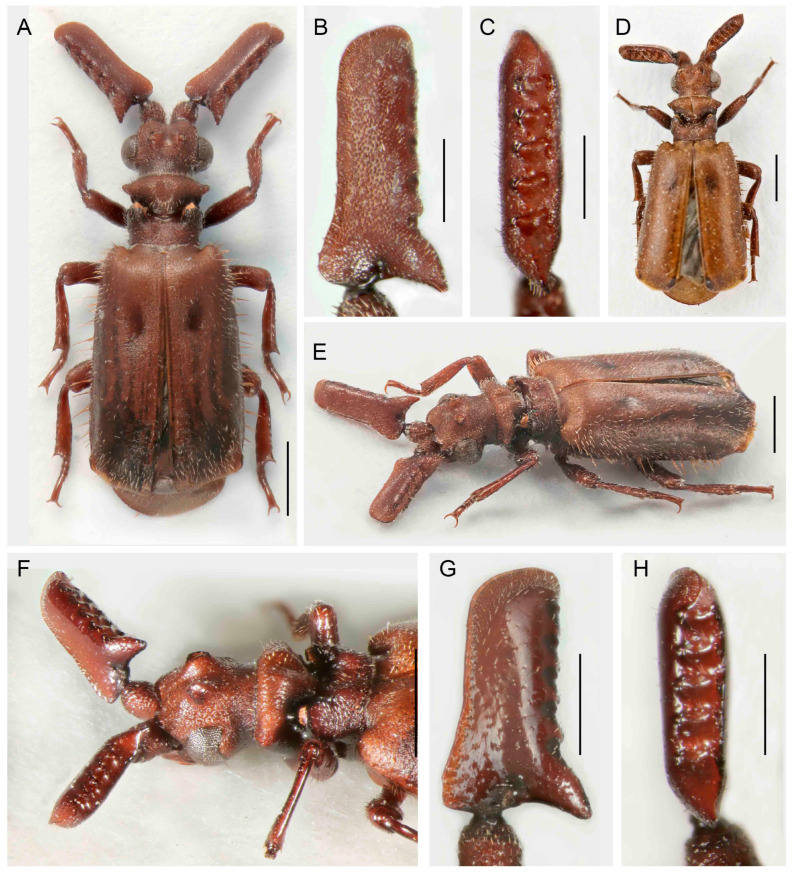
*Paussus mawdsleyi* sp. nov., holotype. (**A**,**E**) Habitus; (**B**,**C**) antennal club in dorsal and lateral views; paratype, female from BMNH: (**D**) habitus; (**F**) head and pronotum; (**G**,**H**) antennal club in dorsal and lateral views. Scales 1 mm (**A**,**D**–**F**), 0.5 mm (**B**,**C**,**G**,**H**).

**Figure 6 insects-14-00947-f006:**
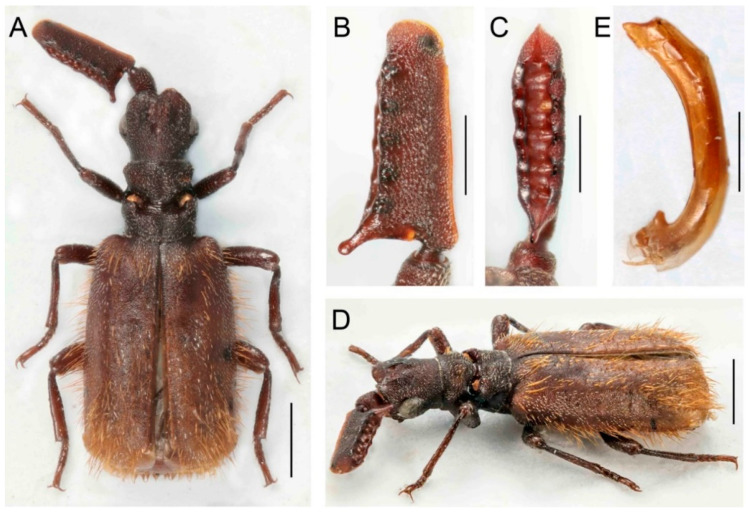
*Paussus bakeri* sp. nov., holotype. (**A**,**D**) Habitus; (**B**,**C**) antennal club in dorsal and lateral views; (**E**) male genitalia. Scales 1 mm (**A**,**D**), 0.5 mm (**B**,**C**,**E**).

**Figure 7 insects-14-00947-f007:**
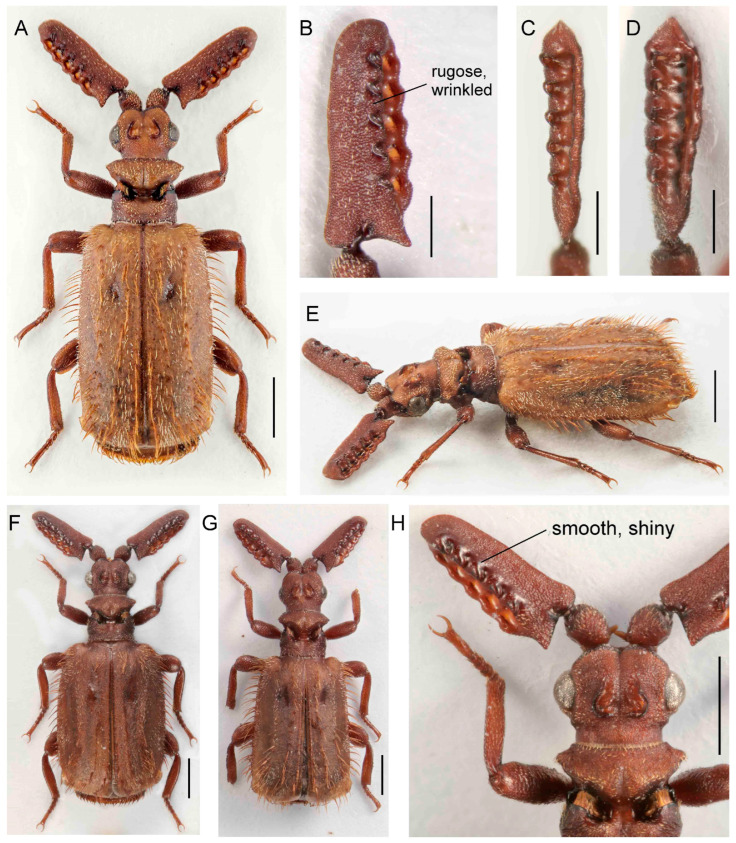
*Paussus jendeki* sp. nov., holotype. (**A**,**E**) Habitus; (**B**,**C**) antennal club in dorsal and lateral views; male *P. hystrix* (Laos, a specimen deposited in MBC): (**D**) Antennal club laterally; (**F**) habitus; (**H**) antennal club and head; female *P. hystrix* (Vietnam, a specimen deposited in ZFMK): (**G**) Habitus. Scales 1 mm (**A**,**E**–**G**), 0.5 mm (**B**–**D**,**H**).

**Figure 8 insects-14-00947-f008:**
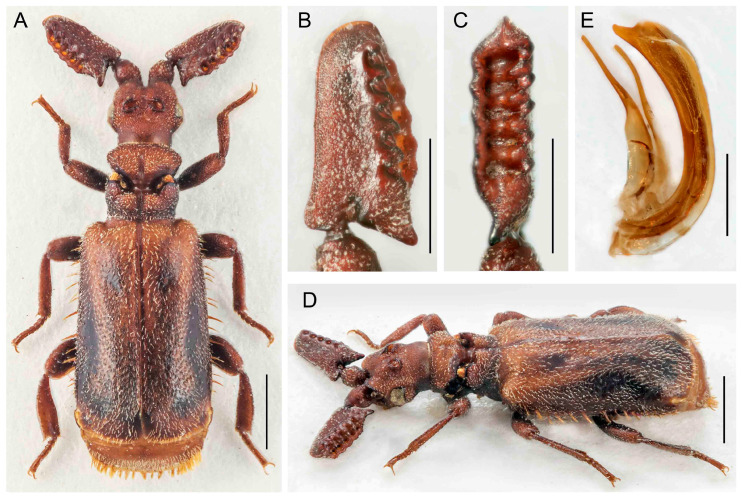
*Paussus saueri* sp. nov., holotype. (**A**,**E**) Habitus; (**B**,**C**) the antennal club in dorsal and lateral views; (**D**) male genitalia. Scales 1 mm (**A**,**E**), 0.5 mm (**B**–**D**).

**Figure 9 insects-14-00947-f009:**
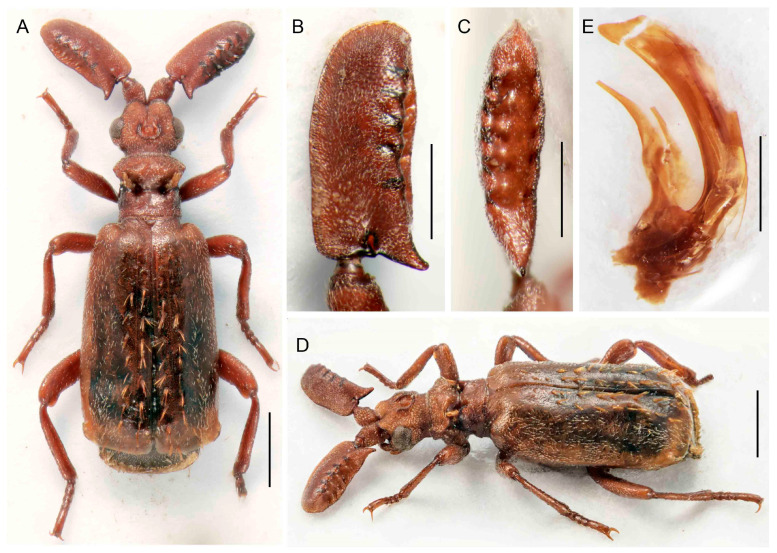
*Paussus annamensis* sp. nov., holotype. (**A**,**D**) Habitus; (**B**,**C**) antennal club in dorsal and lateral views; (**E**) male genitalia. Scales 1 mm (**A**,**D**), 0.5 mm (**B**,**C**,**E**).

**Figure 10 insects-14-00947-f010:**
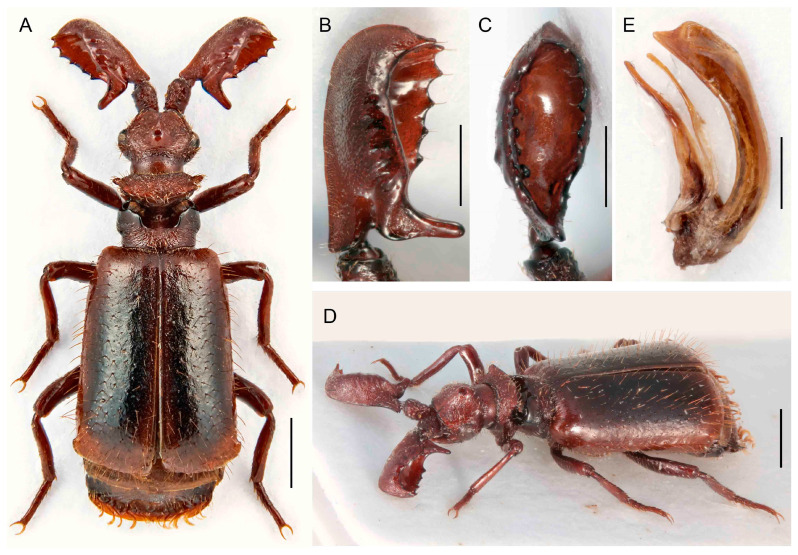
*Paussus phoupanensis* sp. nov., holotype. (**A**,**E**) Habitus; (**B**,**C**) antennal club in dorsal and lateral views; (**D**) male genitalia. Scales 1 mm (**A**,**E**), 0.5 mm (**B**–**D**).

**Figure 11 insects-14-00947-f011:**
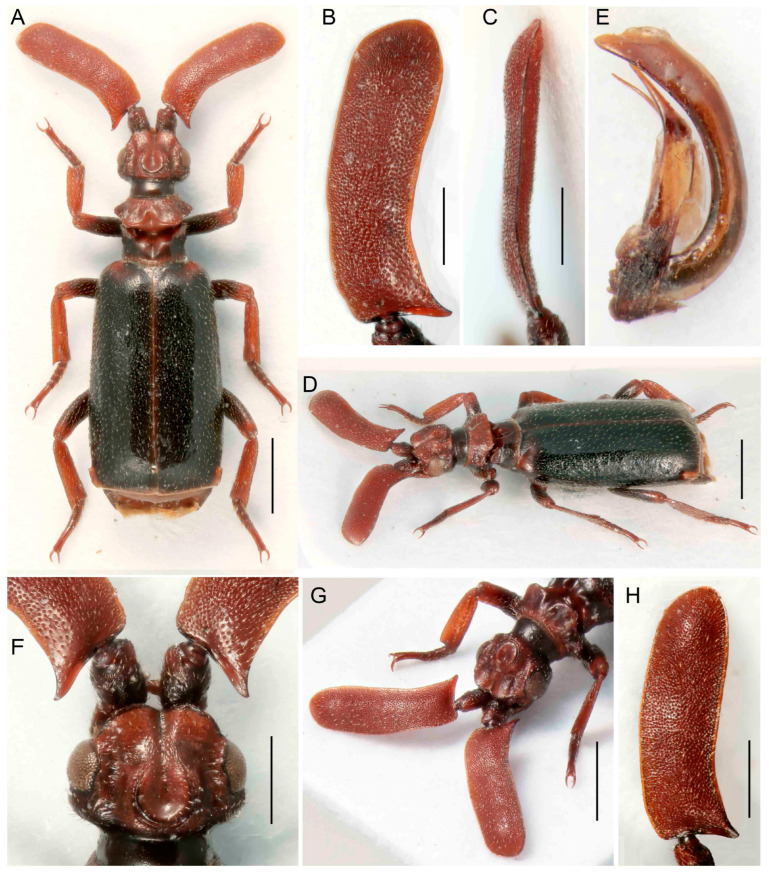
*Paussus bilyi* sp. nov., holotype. (**A**,**D**) Habitus; (**B**,**C**) antennal club in dorsal and lateral views; (**D**) male genitalia; (**F**) head, dorsally with frontal cleft; (**G**) head and antennae dorso-laterally (paratype female); (**H**) antennal club dorsally (paratype female). Scales 1 mm (**A**,**E**,**G**), 0.5 mm (**B**–**D**,**F**,**H**).

**Figure 12 insects-14-00947-f012:**
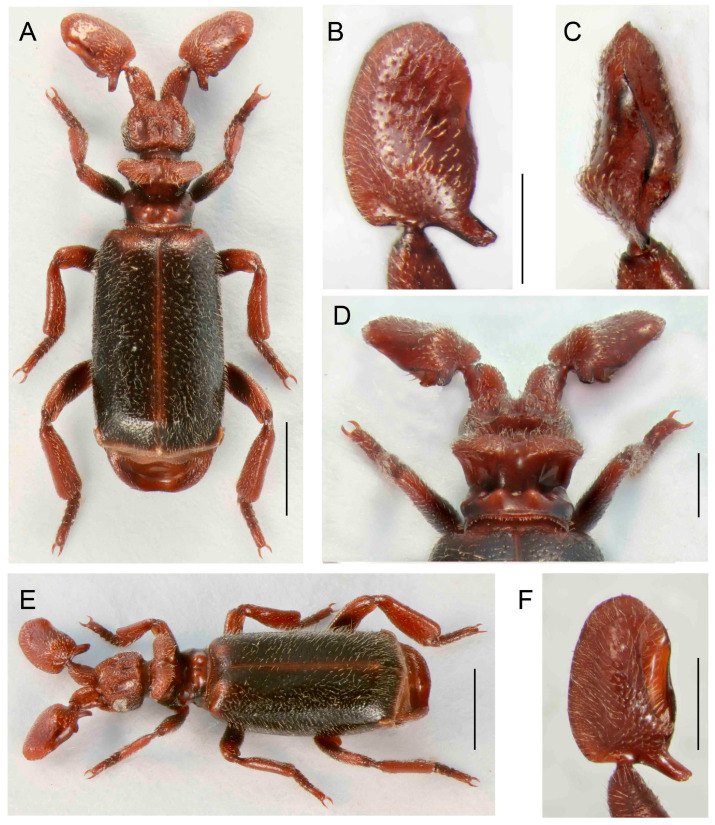
*Paussus haucki* sp. nov., holotype. (**A**,**E**) Habitus; (**B**,**C**) antennal club in dorsal and lateral views; (**D**) head and pronotum, postero-dorsal view; (**F**) antennal club in ventral view. Scales 1 mm (**A**,**E**), 0.5 mm (**B**–**D**,**F**).

**Figure 13 insects-14-00947-f013:**
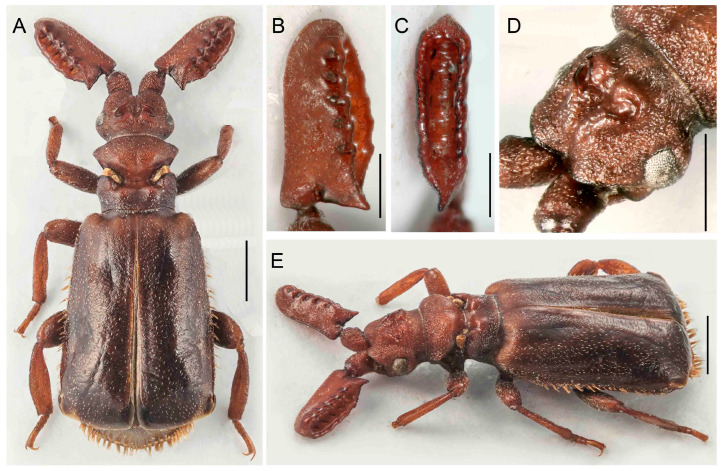
*Paussus madurensis* Wasmann, 1913, female. (**A**,**E**) Habitus; (**B**,**C**) antennal club in dorsal and lateral views; (**D**) frontal crest. Scales 1 mm (**A**,**E**), 0.5 mm (**B**–**D**).

**Figure 14 insects-14-00947-f014:**
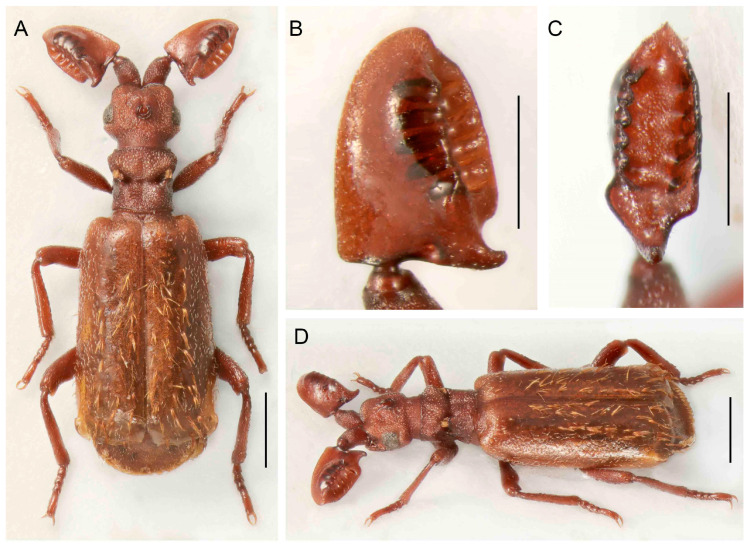
*Paussus corporaali* Reichensperger, 1927, female. (**A**,**D**) Habitus; (**B**,**C**) antennal club in dorsal and lateral views. Scales 1 mm (**A**,**D**), 0.5 mm (**B**–**C**).

## Data Availability

We list all depositories of studied specimens.

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
