# Peer review of "New Species of Paussus, Subgenus Scaphipaussus (Coleoptera: Carabidae: Paussinae), from Southeast Asia Reveal Ambiguities in Species Group Limits and High Species Diversity in the Oriental Region"

_insects, 2023, doi:10.3390/insects14120947_

Round 1

Reviewer 1 Report

Comments and Suggestions for Authors

see attached file

Author Response

Many thanks for the comments and positive review. We cave carefully modified the manuscript as proposed. Our apology for errors.

Reviewer 2 Report

Comments and Suggestions for Authors

This is fine solid taxonomic paper, no any major issue detected expect that a slight and optional improvement of writing style could be suggested. A more major suggestion, but also optional could be to replace lengthy Introduction by a shorter text and an illustration about phylogeny of the genus, informal clades vs formal taxa, and taxonomic characters vs synamomorphies. This diagram would improve the paper and also it would allow to shorten the discussion.

Comments on the Quality of English Language

This is fine solid taxonomic paper, no any major issue detected expect that a slight and optional improvement of writing style could be suggested. A more major suggestion, but also optional could be to replace lengthy Introduction by a shorter text and an illustration about phylogeny of the genus, informal clades vs formal taxa, and taxonomic characters vs synamomorphies. This diagram would improve the paper and also it would allow to shorten the discussion.

Author Response

Reviewer 2

This is fine solid taxonomic paper, no any major issue detected expect that a slight and optional improvement of writing style could be suggested. A more major suggestion, but also optional could be to replace lengthy Introduction by a shorter text and an illustration about phylogeny of the genus, informal clades vs formal taxa, and taxonomic characters vs synamomorphies. This diagram would improve the paper and also it would allow to shorten the discussion.

Thanks for your comment. We added into Figure 1 a diagram showing the earlier and current classification of Paussus subgenera as proposed. The topic discussed in length by Robertson & Moore (2017) and due to very complex situation it is still needed to summarize their fundings in the introduction. The whole Introduction is about 3 pages long and fall within the usual length in similar studies. We discuss classification and distribution and the goal of our study. Only shortly we summarize the classification history of the group.

This is fine solid taxonomic paper, no any major issue detected expect that a slight and optional improvement of writing style could be suggested. A more major suggestion, but also optional could be to replace lengthy Introduction by a shorter text and an illustration about phylogeny of the genus, informal clades vs formal taxa, and taxonomic characters vs synamomorphies. This diagram would improve the paper and also it would allow to shorten the discussion.

The identification of synapomorphies was discussed in Robertson & Moore (2017) and we cannot contribute in the present study to the morphological classification of the whole genus. Therefore, we discuss only Scaphipaussus. Concerning the fact that its delimitation is based on DNA relationships (see the new figure and the text), we cannot identify clear synapomorphies.

The discussion occupies less than two pages of the 40-page manuscript (979 words and 6180 characters). Although we can technically shorten it, it would be at the expense of clarity.

Reviewer 3 Report

Comments and Suggestions for Authors

The study presents a comprehensive investigation into the species diversity of Asian Paussus, with a specific focus on the subgenus Scaphipaussus. Despite the challenges posed by the rarity of these highly modified ground beetles living within ant nests, the research led to the remarkable discovery of nine new species, showcasing their high phenotypic divergence. The use of morphological data aids in the provisional classification of species groups and identification of diversity centers in Southeast Asia, complementing the latest molecular phylogeny-based classification of Paussus. The paper outlines the taxonomic complexity of Paussus, highlighting the description of nine new species from Southeast Asia and China. The inclusion of diverse morphological characteristics, such as antennal morphology, frontal protuberances and other features, provides valuable insights into species-group delimitation. The emphasis on the rapid radiation and increasing species diversity of Scaphipaussus in the last decade underscores the importance of ongoing research in elucidating the evolutionary origins and intricate relationships between these beetles and their ant hosts. Overall, this research significantly contributes to our understanding of the biodiversity and taxonomic history of Paussus, paving the way for future studies and discoveries in this intriguing field.

One notable limitation is the scarcity of specimens, which poses a challenge for obtaining DNA samples. This limitation hinders a more comprehensive molecular analysis that could enhance the accuracy of the classification. Additionally, the provisional use of morphological data, while valuable, may lack the precision provided by molecular techniques.  

One suggestion is to be carefully considered. Is it possible to add a key to the subgenus Scaphipaussus from Southeast Asia? Otherwise, it is a great paper to be accepted for publication.  

Author Response

Reviewer 3

The study presents a comprehensive investigation into the species diversity of Asian Paussus, with a specific focus on the subgenus Scaphipaussus. Despite the challenges posed by the rarity of these highly modified ground beetles living within ant nests, the research led to the remarkable discovery of nine new species, showcasing their high phenotypic divergence. The use of morphological data aids in the provisional classification of species groups and identification of diversity centers in Southeast Asia, complementing the latest molecular phylogeny-based classification of Paussus. The paper outlines the taxonomic complexity of Paussus, highlighting the description of nine new species from Southeast Asia and China. The inclusion of diverse morphological characteristics, such as antennal morphology, frontal protuberances and other features, provides valuable insights into species-group delimitation. The emphasis on the rapid radiation and increasing species diversity of Scaphipaussus in the last decade underscores the importance of ongoing research in elucidating the evolutionary origins and intricate relationships between these beetles and their ant hosts. Overall, this research significantly contributes to our understanding of the biodiversity and taxonomic history of Paussus, paving the way for future studies and discoveries in this intriguing field.

One notable limitation is the scarcity of specimens, which poses a challenge for obtaining DNA samples. This limitation hinders a more comprehensive molecular analysis that could enhance the accuracy of the classification. Additionally, the provisional use of morphological data, while valuable, may lack the precision provided by molecular techniques.  

One suggestion is to be carefully considered. Is it possible to add a key to the subgenus Scaphipaussus from Southeast Asia? Otherwise, it is a great paper to be accepted for publication.   

The preparation of the key for 85 species is very problematic. We do not have access to all species, and some are known from their descriptions only. The recent studies by Maruyama, Wang and Nagel provided excellent illustrations. As we show, the species are well defined by morphology, but construction of the key would probably finish with theses – “antenna like in Fig. xx versus antenna different. The illustrations and distribution enable quick and reliable identification of species. We apologize, but currently, it is impossible to prepare such a complex identification key.